# *Arabidopsis thaliana* Sucrose Phosphate Synthase A2 Affects Carbon Partitioning and Drought Response

**DOI:** 10.3390/biology12050685

**Published:** 2023-05-06

**Authors:** Laura Bagnato, Edoardo Tosato, Libero Gurrieri, Paolo Trost, Giuseppe Forlani, Francesca Sparla

**Affiliations:** 1Department of Pharmacy and Biotechnology FaBiT, University of Bologna, 40126 Bologna, Italy; laura.bagnato2@unibo.it (L.B.); edoardo.tosato2@unibo.it (E.T.); libero.gurrieri2@unibo.it (L.G.); paolo.trost@unibo.it (P.T.); 2Department of Life Science and Biotechnology, University of Ferrara, 44121 Ferrara, Italy; flg@unife.it

**Keywords:** osmotic stress, sucrose phosphate synthase, glucose-6-phosphate dehydrogenase, fructose, oxidative pentose phosphate pathway

## Abstract

**Simple Summary:**

Sucrose phosphate synthases catalyze the rate-limiting step in the sucrose biosynthetic pathway. Considering the essential role of sucrose in plant metabolism and molecular signaling, sucrose phosphate synthases are finely tuned both at transcriptional and post-translational levels. In plants, sucrose phosphate synthases are organized in protein families. Arabidopsis contains four sucrose phosphate synthase isoforms, each with a specific but overlapping expression profile, although the literature shows some discrepancies. In the present study, the role of isoform A2 was investigated under control conditions and in response to drought stress. The lack of sucrose phosphate synthase A2 affects (i) carbon partitioning through the activation of the oxidative pentose phosphate pathway and (ii) influences plant response to drought.

**Abstract:**

Sucrose is essential for plants for several reasons: It is a source of energy, a signaling molecule, and a source of carbon skeletons. Sucrose phosphate synthase (SPS) catalyzes the conversion of uridine diphosphate glucose and fructose-6-phosphate to sucrose-6-phosphate, which is rapidly dephosphorylated by sucrose phosphatase. SPS is critical in the accumulation of sucrose because it catalyzes an irreversible reaction. In *Arabidopsis thaliana*, SPSs form a gene family of four members, whose specific functions are not clear yet. In the present work, the role of SPSA2 was investigated in Arabidopsis under both control and drought stress conditions. In seeds and seedlings, major phenotypic traits were not different in wild-type compared with *spsa2* knockout plants. By contrast, 35-day-old plants showed some differences in metabolites and enzyme activities even under control conditions. In response to drought, *SPSA2* was transcriptionally activated, and the divergences between the two genotypes were higher, with *spsa2* showing reduced proline accumulation and increased lipid peroxidation. Total soluble sugars and fructose concentrations were about halved compared with wild-type plants, and the plastid component of the oxidative pentose phosphate pathway was activated. Unlike previous reports, our results support the involvement of SPSA2 in both carbon partitioning and drought response.

## 1. Introduction

Sucrose (Suc) is a disaccharide synthesized only by oxygenic photosynthetic organisms [1,2]. In plants, Suc and starch are the main end products of CO_2_ fixation occurring during the day. In leaves, photosynthetic triose phosphate production fuels either Suc synthesis in the cytosol or transitory starch synthesis in the chloroplast stroma [3]. During the subsequent night, transitory starch is degraded, and soluble sugars, mainly maltose, are exported to the cytosol to sustain Suc synthesis [4,5,6]. In many plants, Suc is also the main transported sugar, exported from source tissues and imported in sink ones [7]. Besides acting as a mobile energy source, Suc can also serve as a signaling molecule or as a source of carbon skeletons, and it is often accumulated in response to environmental stresses [8,9,10,11,12]. 

The pool of hexoses sustains Suc biosynthesis: Fructose-6-phosphate (F6P) and uridine diphosphate glucose (UDP-Glc) are the substrates of Suc-phosphate synthase (SPS; EC 2.4.1.14) that catalyzes the production of Suc-6-phosphate (S6P), which is hydrolyzed in turn to Suc by Suc-phosphate phosphatase (SPP; EC 3.1.3.24). The limiting step in the biosynthesis of Suc is the production of S6P, and SPS is the main regulator of Suc synthesis [13]. Several mechanisms of regulation tune SPS activity. Glucose-6-phosphate (G6P) and inorganic phosphate (P_i_) act as allosteric regulators, the former being an activator and the latter, an inhibitor [13]. Allosteric regulation also depends on the post-translational modification that occurs in the dark via the phosphorylation of a specific Serine residue, which makes P_i_ inhibition stronger [13,14]. 

The Arabidopsis genome encodes four isoforms of SPS: At5g20280, named *SPSA1* (or *SPS1F*); At5g11110 named *SPSA2* (or *SPS2F*); At1g04920 named *SPSB* (also known as *SPS3F*); and At4g10120 named *SPSC* (also known as *SPS4F*). All SPS isoforms of Arabidopsis are catalytically active when expressed in yeast, although the specific activity and allosteric regulation by G6P and P_i_ markedly vary among isoforms [15].

Comparative studies of *SPS* gene expression conducted through the analysis of the promoter regions revealed a partially overlapping expression pattern within the organs and tissues of Arabidopsis plants, with some discordant data among the different experiments [15,16,17,18]. In agreement with the overlapping expression profiles, *SPS* single mutants are not affected in their growth and reproduction [15,16]. In Arabidopsis, SPSA1 and SPSC have been reported as the main isoforms, and the effects of the deletion of one of the two genes are compensated by the presence of the other, guaranteeing a normal plant growth and normal accumulation of starch and sucrose [15,16]. Conversely, the double-mutant *spsa1/spsc* showed altered plant morphology and alterations in the biosynthesis and export of sucrose, both under light and dark conditions [17]. The addition of the third mutation in *SPSB* gave rise to sterile plants, while the quadruple knockout mutant was lethal [15,17]. 

The oxidative pentose phosphate pathway (OPPP) is a major source of reducing power and metabolic intermediates for biosynthetic processes. G6P is the starting point of the OPPP. Through three reactions catalyzed by glucose-6-phosphate dehydrogenase (G6PD, EC 1.1.1.49), 6-phosphogluconolactonase (6PGL, EC 3.1.1.31), and 6-phosphogluconate dehydrogenase (6PGDH, EC 1.1.1.44), G6P undergoes an oxidative decarboxylation, providing reducing power in the form of two molecules of NADPH and resulting in one molecule of ribulose-5-phosphate (Ru5P) with the release of one CO_2_. The reducing power provided by the two dehydrogenases G6PD and 6PGDH significantly contributes to the anabolic pathways not only in non-photosynthetic tissues but also in photosynthetic ones in the dark and in response to stress [19,20]. Furthermore, the NADPH produced by the above-mentioned enzymes plays an essential role in maintaining cellular redox homeostasis by providing reducing equivalents to glutathione reductase, thus keeping the glutathione pool reduced [21,22]. 

In plants, the OPPP occurs in both the cytosol and plastids, and the whole cycle is strictly controlled by the first enzyme of the cycle, G6PD, which is subject to feedback inhibition by NAPDH [23,24]. In addition, plastid isoforms are further regulated by the thioredoxin–ferredoxin system [25,26].

The plastid isoforms of G6PD are less sensitive to NADPH-mediated feedback inhibition and are divided into two classes based on the type of plastid: G6PDs of chloroplasts belong to the P1 class, while G6PD of heterotrophic cells belong to the P2 class [27]. Both P1 and P2 types increase their activity upon oxidation. The thioredoxin-dependent inhibition of P1-type avoids the futile cycle that would be established in the light around G6P, derived from the metabolic phase of photosynthesis, and NADPH, produced by the photosynthetic electron transport chain. By contrast, the dark activation of the P1 type provides NADPH and phosphorylated sugars via the OPPP for stromal biosynthesis and redox balance [25]. In non-photosynthetic plastids, the NADPH produced by the P2 type is almost entirely consumed in the reduction of ferredoxin, which in turn is required for the reductive steps in nitrite and sulfate metabolism [28]. 

The Arabidopsis genome contains multiple genes encoding G6PD: Two genes encode cytosolic G6PDs (isoforms 5 and 6), three genes encode plastid-located G6PDs (isoforms 1, 2, and 3) and a sixth gene encodes the non-functional isoform 4, also named the P0 type [29]. Although devoid of catalytic activity, G6PD4 retains the two cysteines involved in redox regulation and required to interact with G6PD1, with the resulting heterodimer G6PD1/G6PD4 being imported into the peroxisomes [26,27,30,31].

In the present work, the role of SPSA2 in the absence of and in response to osmotic stress was studied. In agreement with previous data, *SPSA2* was induced by osmotic stress [18], emphasizing its involvement in the stress response, but interestingly, the lack of *SPSA2* affected carbon partitioning in adult plants, even under physiological conditions. In the absence of SPSA2, carbon skeletons were apparently metabolized through the OPPP, which was found to be more active in these plants than wild-type plants. This activation correlated with the downregulation of the catalytically inactive isoform 4 of G6PD, strongly suggesting a negative regulatory function of G6PD4.

## 2. Materials and Methods

### 2.1. Plant Materials

A T-DNA line (SALK_064922C) for the At5g11110 gene was purchased from the European Arabidopsis Stock Centre (NASC, Nottingham, UK). The homozygous T-DNA insertion was confirmed via PCR on 400 ng of genomic DNA extracted from the leaves of mutated plants. T-DNA specific primer LBb1.3 and gene-specific primers *SPSA2*-RP and *SPSA2*-LP (Appendix A) were used in combination (*SPSA2*-LP + *SPSA2*-RP and LBb1.3 + *SPSA2*-RP) for two independent PCR reactions. For comparison, PCR reactions were also conducted under the same conditions on genomic DNA extracted from wild-type plants. A Biometra T-gradient thermocycler was used for PCR amplifications with *Taq*-polymerase (NEB), following the manufacturer’s instructions. PCR products were visualized on 0.8% agarose gel containing 1X GelRed (Merck Life Science S.r.l., Milano, Italy).

The residual expression level of *SPSA2* in the homozygous line was assessed using RT-PCR on the total RNA extracted from wild-type and *spsa2* plants. Total RNA isolation was performed using a SpectrumTM Plant Total RNA Kit (Merck) on two independent biological replicates for each genotype. RNA concentration and purity were determined using a NanoDrop^®^ spectrophotometer ND-1000 (Thermo Fisher Scientific, Waltham, MA, USA) and controlled on 1% agarose gel. RNA samples were treated with RNase-free DNase I (Thermo Fisher Scientific, Waltham, MA, USA), and then single-stranded cDNA was generated using an EasyScript^®^ First-Strand cDNA Synthesis Super Mix (Trans^®^), following the manufacturer’s instructions. Two independent PCR reactions were performed using *SPSA2*-specific primers and *PP2A* (At2g13320)-specific primers (Appendix A). PCR products were visualized on 0.8% agarose gel. The intensity of the bands was quantified with the free and open-source software package ImageJ and *SPSA2* signals were normalized on the corresponding *PP2A* bands.

### 2.2. Plant Growth Conditions

*Arabidopsis thaliana* wild-type and knockout plants were grown on soil, in a hydroponic solution, or on Petri dishes. For plants grown on soil, after 2/3 days of stratification, pots containing 4 plants each were transferred to a growth chamber at a constant temperature of 22 °C under a 12/12 h light/dark cycle and with a photosynthetic photon flux density of 100–120 µmol m^−2^ s^−1^. The plants were watered as needed. 

For hydroponic cultures, surface-sterilized seeds were placed on seed holders filled with 0.65% (*w*/*v*) agarose and lodged into the lid of a black polyethylene box containing a sterile hydroponic solution (1.25 mM KNO_3_; 1.5 mM Ca(NO_3_)_2_; 0.75 mM MgSO_4_; 0.5 mM KH_2_PO_4_; 50 µM Fe(II)-EDTA; 50 µM H_3_BO_3_; 1 µM ZnSO_4_; 0.7 µM CuSO_4_; 12 µM MnSO_4_; 0.24 µM Na_2_MoO_4_; and 0.1 mM Na_2_SiO_3_). Boxes were incubated at 4 °C for 2/3 days and then transferred to a growth chamber under a 12/12 h light/dark cycle, with a photosynthetic photon flux density of 100–120 µmol m^−2^ s^−1^ and at a constant temperature of 22 °C. Osmotic stress was applied on 35-day-old plants at 12 h dark, changing the hydroponic medium with a fresh one supplemented with 150 mM mannitol. The plants were collected at the end of the light period and at different time points.

For the plants grown on Petri dishes, a half-strength Murashige and Skoog (½MS) agar medium was used to sow the sterilized seeds. Drought stress was applied by adding different concentrations of mannitol (50, 100, and 150 mM mannitol) in the ½MS medium. After 2/3 days of stratification at 4 °C, the plants were transferred at room temperature (about 19 °C) under a 12/12 h light/dark cycle and a photosynthetic photon flux density of 100–120 µmol m^−2^ s^−1^.

### 2.3. Expression Levels of SPSA2 in Response to Drought

The expression level of *SPSA2* in response to drought was detected with quantitative PCR (qPCR) performed on a CFX Connect™ Real-Time PCR Detection System (Bio-Rad Laboratories Inc., Hercules, CA, USA) on the total RNA extracted from wild-type plants. Total RNA extraction and cDNA synthesis were performed as described above; for cDNA synthesis, 1 μg of RNA was used as the template. qPCR was performed in an optical 96-well plate with 10 μL of a reaction containing 1.5 μL of cDNA, 400 nM of specific primers, and SYBR^®^ Green JumpStart™ *Taq* ReadyMix (Merck) to monitor double-strand DNA synthesis. Reactions were performed in triplicates for each gene and on three samples for each time point. The following cycling profile was used: 2 min at 95 °C followed by 40 cycles of 5 s of denaturation (95 °C) and 30 s of elongation at 60 °C.

The relative expression was calculated following the ΔCt method [32] and normalized on the housekeeping gene *ACT2* (At3g18780). For each cDNA sample, the average Ct (threshold cycle) of the reference gene was subtracted from the average Ct of *SPSA2*. The relative expression was obtained by calculating 2^ΔCt^ and normalizing the expression with respect to CTR samples. *SPSA2* and *ACT2* primer sequences are reported in Appendix A. Primer efficiency was calculated by performing quantitative real-time PCR on four serial dilutions starting from 200 ng of genomic DNA, and Ct values were plotted and fitted linearly. The slope was used to obtain the efficiency.

### 2.4. Seed Permeability, Seed Viability, and Seed Germination

Seed permeability and mucilage formation were tested through 2,3,5-triphenyl-tetrazolium chloride (TZ) assay and ruthenium red (RR) staining, respectively, as described in [33].

To determine the percentage of germination, the sterilized seeds were plated on a square Petri dish (10 × 10 cm) filled with the ½MS agar medium in the absence or presence of mannitol (50, 100, and 150 mM mannitol). Each Petri dish contained about 100 seeds, sown by alternating 2 rows of wild-type seeds with 2 rows of *spsa2* seeds. The percentage of germination was calculated by counting the plantlets at 7 and 13 days after stratification. Approximately 100 seeds for each genotype were scored.

### 2.5. Qualitative Evaluation of Reactive Oxygen Species and Detection of Lipid Peroxidation

The accumulation of reactive oxygen species (ROS) in Arabidopsis seedlings was detected using nitroblue tetrazolium (NBT) staining. Briefly, 2-week-old plants were transferred in a 50 mM K phosphate buffer, pH 7.5, in the presence of 0.5% Triton X-100 and 0.01% NBT. The plants were incubated at room temperature for 1 h under mild shaking. After incubation, the plants were transferred to 80% ethanol until bleaching. The qualitative assessment of NBT staining was performed on 30 plants by 7 different operators. Seedlings were clustered in two classes: class I, corresponding to lowly stressed plants, and class II, corresponding to highly stressed plants.

Lipid peroxidation in adult plants exposed to drought was assessed by 2-thiobarbituric acid (TBA) assay, as described in [34]. The malondialdehyde (MDA)–TBA adduct was monitored at 532 nm, subtracting the non-specific absorption at 600 nm. 

### 2.6. Determination of Water Content

Water content was determined on 25 adult plants for each genotype and stress condition. The plants were harvested at 12 h light and immediately weighed to determine the fresh weight (FW). Excised rosettes were then dried at 80 °C for 24 h and weighed again for the determination of the dry weight (DW). Water content was calculated as the difference between the FW and DW of each plant and expressed as a percentage of the FW.

### 2.7. Quantification of Carbohydrates, Proline, and Amino Acid Pools

Starch quantification was performed on the entire rosettes of plants collected at 12 h light at different times following mannitol treatment. For each time point, three independent biological samples were measured. Each sample consisted of 3 to 6 rosettes, pooled together and pulverized with liquid nitrogen. Leaf starch content was measured on powdered samples as described in [35]. Lugol’s solution was used for the qualitative detection of leaf starch content in the Arabidopsis plants collected as above. 

The total water-soluble carbohydrates (WSCs) were extracted from Arabidopsis powders with cold MilliQ^®^ water and then quantified by means of an Anthrone assay, as described in [36]. The analysis of reducing soluble sugars was conducted with a 3,5-dinitrosalicylic acid (DNS) assay, as described in [37]. The quantification of glucose and fructose contents was performed using a Sucrose/D-Fructose/D-Glucose Assay Kit (Megazyme International Ireland, Co. Wicklow, Ireland), following the manufacturer’s instructions. 

Cell wall carbohydrates, proline (Pro), and amino acid content were quantified as described in [36].

For all quantification assays, data are reported as means ± standard deviations (SDs).

### 2.8. Enzyme Assays

G6PD and phosphofructokinase (PFK) activities were measured on the soluble fraction of the total leaf extracts obtained after 20 min centrifugation at 13.000× *g*. Soluble proteins were extracted from leaf powders by vortexing, three times for 1 min each, a mixture obtained by suspending the leaf powder and extraction buffer in a 1:3 ratio. Extraction buffer for G6PD activity assay was: 30 mM Hepes-Tris, pH 7.8; 3 mM MgCl_2_, 1 mM EDTA, 1 mM phenyl methyl sulfonyl fluoride (PMSF). Extraction buffer for PFK activity assay was: 50 mM Hepes-KOH, pH 6.8, 15% glycerol, 5 mM Mg-Acetate, 1 mM EDTA, 5 mM DTT, 0.1 mM PMSF. The protein concentration of the samples was determined using a bicinchoninic acid (BCA) assay. G6PD and PFK activities are reported as a function of protein content.

G6PD activity was measured following the reduction of NADP^+^ to NADPH at 340 nm. Briefly, after 30 min incubation at 22 °C in the presence of 20 mM of reduced or oxidized DTT, 35 µL of the protein sample was transferred into a cuvette in the presence of 30 mM Hepes-Tris, pH 7.8; 3 mM MgCl_2_; 1 mM EDTA; and 0.5 mM NADP^+^. The blank reaction was monitored for nearly 1 min before starting the reaction with the addition of 5 mM glucose 6-phosphate [38]. 

PFK activity was measured following the oxidation of NADH to NAD^+^ at 340 nm. Briefly, the protein sample was transferred to a cuvette in the presence of 100 mM Hepes-KOH, pH 7.9; 2 mM MgCl_2_; 0.15 mM NADH; 7.5 mM fructose 6-phosphate; 1 mM reduced DTT; 1 U aldolase; 1 U triosephosphate isomerase; and 1 U glycerol 3-phosphate dehydrogenase. The reaction was monitored for about 1 min (blank) before the addition of 2.5 mM ATP [39].

### 2.9. Expression Levels of G6PD Isoforms

The expression levels of *G6PD* isoforms were detected with RT-PCR on the total RNA extracted from wild-type and *spsa2* plants. Total RNA extraction and single-stranded cDNA generation were performed as described in Section 2.1. RT-PCR reactions were performed using *G6PD*-specific and *PP2A*-specific primers (Appendix A), following the manufacturer’s instructions. PCR products were visualized on 0.8% agarose gel. The intensity of the bands was quantified with ImageJ, and *SPSA2* signals were normalized on the proper *PP2A* bands.

### 2.10. Quantification of NADP(H) in Leaf Extracts

NADP(H) levels were measured following the method of [40]. Plant material (10–12 rosettes) was powdered in liquid nitrogen, and the power was divided in two and suspended in 2 mL g^−1^ of either 0.2 M HCl or 0.2 M NaOH. The homogenate was centrifuged for 3 min at 12,000× *g* at RT. Aliquots (240 μL) of the supernatant were treated at 98 °C for 3 min and then neutralized via the addition of 24 μL of 200 mM Na phosphate buffer, pH 5.6, and a proper volume of 0.2 M NaOH or HCl in a final volume of 480 μL, so as to obtain a pH between 6 and 7. Following further centrifugation as above, 7 sample aliquots (from 5 to 20 μL) were placed in wells of a microtiter plate in a final volume of 200 μL containing 0.05 M Hepes-NaOH (pH 7.5), 1 mM EDTA, 0.12 mM dichlorophenol indophenol, 1 mM phenazine methosulphate, and 0.5 mM glucose 6-phosphate. The reaction was started through the addition of 0.02 μL G6PD suspension (Sigma G7877). The resulting decrease in absorbance at 595 nm was monitored at 20 s intervals for 5 min, and rates were calculated over the linear part of the patterns using a calibration curve obtained with known amounts (200–600 pmol) of an authentic NADP(H) standard. Reduced and oxidized pyridine dinucleotides were distinguished by preferential destruction in acid or base, respectively.

## 3. Results

### 3.1. Selection of Knockout spsa2 Mutant

A T-DNA insertion line for the *SPSA2* gene (At5g11110) (Appendix A) was searched in TAIR’s database, and the corresponding seed stock (SALK_064922C) was purchased from the European Arabidopsis Stock Centre (NASC, Nottingham, UK). Homozygous T-DNA individuals were selected using PCR on the genomic DNA extracted from wild-type and T-DNA plants grown on soil (Appendix A). The T-DNA insertion site was initially assigned to the fifth exon (Genome Assembly TAIR9) and then to the fifth intron (Genome Assembly TAIR10). The effect of the insertion was thus assessed using RT-PCR on the total RNA extracted from adult homozygous plants grown on soil, and the absence of the *SPSA2* signal confirmed *spsa2* as a knockout (KO) mutant line (Appendix A).

### 3.2. Under Tested Conditions, Seeds and Seedlings Were Not Affected by the Absence of SPSA2

To test the possible role of SPSA2 in seeds and seedlings, the permeability of the seed coat and the formation of the mucilage capsule upon seed hydration were evaluated using TZ and RR assays, respectively. As shown in Figure 1a, neither assay highlighted any difference between wild-type and KO plants, suggesting a marginal or null role of SPSA2 in seed viability. 

Similarly, the percentage of germination was not affected by the absence of SPSA2. As expected, the addition of mannitol in the range of 50 to 150 mM slightly decreased the percentage of germination, with no differential effects on the two genotypes (Figure 1b). In agreement with the similar germination rate, NBT staining showed an increase in reactive oxygen species (ROS) in response to drought conditions. However, no statistically significant differences were observed between KO and wild-type seedlings under control or stress conditions (Appendix A).

### 3.3. Osmotic Stress Induced the Expression of SPSA2

The Arabidopsis genome encodes four SPS isoforms. The expression of the four SPS isoforms analyzed in silico using the eFP browser showed a nearly 14-fold specific induction of *SPSA2* after 24 h exposure to osmotic stress induced by 300 mM mannitol (Appendix A) [41]. To verify the induction of *SPSA2* even under mild osmotic stress, 35-day-old wild-type plants, grown hydroponically, were transferred into a medium containing 150 mM mannitol. The *SPSA2* expression level was monitored over time at 12 h light on different days after treatment (DAT). As shown in Appendix A, the expression of *SPSA2* was induced by stress conditions.

### 3.4. Water Content and Carbohydrate Pools

Before proceeding with the analysis of other quantitative phenotypic traits (see below), the water content in both wild-type and *spaA2* plants was evaluated under both control conditions and 150 mM mannitol treatment (Appendix A). Since both genotypes were found to lose water at similar rates, the quantitative determination of metabolites and enzyme activities could be expressed and compared between genotypes based on the FW. This avoids drying plants, a procedure that complicates subsequent extraction protocols.

At first, carbohydrate pools were considered. The total water-soluble carbohydrates (WSCs) were quantified using an Anthrone test (Figure 2a), while a DNS assay was performed to evaluate the concentration of the reducing components in WSCs (Figure 2b). In wild-type and *spsa2* plants, most of the quantified WSCs were reducing sugars, at least within the detection limit of the two methods.

Even under control conditions, the *spsa2* mutant showed a lower WSC content than the wild-type sample. The differences increased in the presence of 150 mM mannitol, a condition in which the WSC content in the mutant was approximately halved compared with the wild-type sample (Figure 2a,b). By measuring glucose and fructose in leaf samples, we found that only fructose was markedly lower in *spsa2* than in wild-type plants (Figure 2c,d). The lower fructose content of *spsa2* plants was also observed in control conditions (*spsa2* showed a 42% reduction in fructose content compared with the wild-type sample). At the onset of drought stress (0.5 DAT), the difference in fructose concentration between wild-type and *spa2* plants disappeared and became clearly evident starting from 2.5 DAT (Figure 2d).

Compared with wild-type plants, at the onset of the stress treatment, transitory starch measured at 12 h light was lower in the mutant than in wild-type plants but returned similar values in both genotypes at longer times of stress exposure (4.5 and 6.5 DAT) (Figure 3a and Appendix A). By contrast, the pool of cell wall (CW) carbohydrates was not affected in *spsa2* under control or stress conditions (Figure 3b).

### 3.5. Proline, Amino Acid Pools, and Oxidative Damage

Mannitol stress induces a type of oxidative damage [34,36] that can be quantified using a TBA assay. Under control conditions and up to 2.5 DAT, wild-type and *spsa2* plants showed no difference in oxidative damage (Figure 4). However, at 4.5 and 6.5 DAT, the two genotypes started to diverge, showing greater oxidation in the mutant than in wild-type plants (Figure 4).

Pro is a proteinogenic amino acid known to be a compatible osmolyte that accumulates in response to various stresses, including drought [42]. The Pro concentration in rosette leaves was quantified in both genotypes under control and stress conditions. As expected, both genotypes accumulated Pro in response to drought. However, at 4.5 DAT, *spsa2* accumulated Pro to a lesser extent than wild-type plants (Figure 5a), in agreement with the higher oxidative damage shown by the mutant (Figure 4).

Differently from Pro, the concentration of free amino acids, measured in control and stressed plants, did not display statistically significant differences between mutant and wild-type plants (Figure 5b).

### 3.6. Analysis of PFK and G6PD Activities

With the aim of elucidating the reason for the reduced fructose content measured in *spsa2*, PFK activity in whole-leaf extracts was assayed in wild-type and KO plants under both control and stress conditions, but only small differences were observed between the two genotypes and the two growth conditions (Figure 6a).

Similarly, the total activity of G6PD in leaves was evaluated under both oxidizing (Figure 6b) and reducing (Figure 6c) conditions; the former is known to activate G6PD activity, while the latter is known to inhibit it [25,43,44]. As shown in Figure 6b, under oxidized conditions, G6PD activity was on average 1.6-fold higher in *spsa2* than in wild-type plants from 0 to 4.5 DAT. By contrast, no significant differences were observed between the genotypes, at any time point considered, under reducing conditions (Figure 6c).

### 3.7. Analysis of the Expression Levels of G6PD Isoforms

In Arabidopsis, *G6PD*s form a gene family consisting of six members [29]. In order to understand the basis for the higher G6PD activity in *spsa2*, the expression of all *G6PD* genes was measured using RT-PCR in *spsa2* mutant and wild-type plants under both control conditions and 150 mM mannitol treatment (Figure 7). None of the isoforms showed higher transcript levels in KO plants, except for *G6PD4*, which was clearly less expressed in *spsa2* mutant than in wild-type plants. The lower expression of *G6PD4* in the mutant was also evident in the absence of stress, while at 6.5 DAT, the wild-type plants also had lower *G6PD4* expression levels such that the two genotypes resembled each other.

### 3.8. Comparison of the NADP^+^:NADPH Ratio in spsa2 and Wild-Type Plants

Through the OPPP, G6PD activity provides reducing power in the form of NADPH. Concentrations of NADP(H) were measured in wild-type and *spsa2* plants under both control and mannitol treatment. The NADP(H) concentrations in both genotypes were statistically different only under control conditions, while under stress conditions, no statistically significant differences were observed (Figure 8).

Accordingly, the NADP^+^:NADPH ratio was clearly lower in *spsa2* than in wild-type plants (Table 1). In agreement with that observed for G6PD activity (Figure 6), the difference was visible under control conditions but tended to decrease with the length of the treatment (Table 1).

## 4. Discussion

In Arabidopsis, *SPS* genes form a small family composed of four members, all coding active enzymes [15]. SPSs do not catalyze the production of Suc directly; nonetheless, SPSs are the main regulators of the Suc biosynthetic pathway [13]. Both gene expression and enzyme regulation contribute to tuning the level of Suc in the different plant tissues and developmental stages. Because of the relevant role of Suc as a molecule at the crossroads between different functions (physiological roles of Suc range from signaling molecule to energy source), SPSs have been the subject of several studies. In particular, studies that dissected the promoter regions analyzed the effects caused by the absence of every single gene and their combinations and characterized the specific activity and the response to allosteric effectors exhibited by each isoform expressed in yeast [15,16,17,18]. Nonetheless, some questions on the contribution of individual SPS isoforms still remain open and need further investigation, also because of some inconsistencies in the literature.

In the present study, the physiological effects of the loss of a functional *SPSA2* gene were investigated. We chose to analyze the *spsa2* mutant because *SPSA2* expression was the most controversial, with some data pointing to the null expression of *SPSA2* in mature leaves and others showing the opposite pattern [15,18]. In addition, the promoter region of *SPSA2* was positively regulated by osmotic stress [18]. Considering these observations, we assumed that combining the absence of *SPSA2* with mild osmotic stress could have revealed additional information about its physiological role.

As far as the analyzed traits are concerned, no effect was observed in *spsa2* seeds and seedlings. The permeability of seed coats, mucilage capsule formation, and germination rates were not different between wild-type and mutant plants, even when germination occurred in the presence of mannitol (Figure 1). These data confirm the marginal or null function of SPSA2 in seeds and seedlings, in agreement with [18], who demonstrated that the promoter region of *SPSC* drives the expression in embryos, suggesting that SPSC is the major isoform involved in the early stages of plant development. Nevertheless, we cannot exclude that the absence of a phenotype could be due to the overlapping functions of *SPS* genes [15,17]. At least under the tested conditions and for the reported traits never described before, the mutation did not affect plants.

Contrary to seeds, and unlike what was previously reported [16], in adult plants hydroponically grown in a 12 h light/dark regime, small differences between mutant and wild-type plants were appreciable also in the absence of stress. Under control conditions, the absence of SPSA2 did not increase lipid peroxidation (Figure 4). However, the total soluble sugars, as well as glucose and fructose concentrations, were lower in the mutant than in wild-type plants even in the absence of stress (Figure 2), while starch and cell wall carbohydrates showed no significant differences (Figure 3). In the attempt to pinpoint the pathway through which hexoses were presumably metabolized in the KO plant, the activity of PFK and G6PD, two key enzymes of glycolysis and the OPPP, respectively, were measured (Figure 6). The results clearly showed that only the redox-regulated component of G6PD activity increased in the mutant, even under control conditions, suggesting that the plastid-located isoforms of G6PD (i.e., G6PD1, G6PD2, and G6PD3) [26,27,29] were mainly involved. Interestingly, the increase in the redox-regulated component of G6PD activity did not depend on transcriptional gene activation, since in the absence of stress, no differences were observed between *spsa2* and wild-type plants in the transcriptional levels of *G6PD1*, *G6PD2*, *G6PD3*, *G6PD5*, and *G6PD6*. The increase in redox-regulated G6PD activity conversely correlated with the lower expression of the non-catalytically active isoform 4 (Figure 7). In *Arabidopsis thaliana*, the interaction between G6PD4 and G6PD1 has been demonstrated, and it has been observed that this interaction drives the import of G6PD4/G6PD1 heterodimers in peroxisomes [31]. In an attempt to explain the current results, we suggest that G6PD4 may act as a negative regulator of the redox-sensitive component of G6PDs, possibly by directing G6PD1 in peroxisomes. Further studies are required to elucidate how the peroxisome branch of the OPPP impacts NADPH production.

In agreement with the activation of plastid G6PDs in *spsa2* under control conditions, the concentration of NADPH in *spsa2* was approximately double the concentration of wild-type plants (Figure 8), with a NADP^+^:NADPH ratio of about 1.5-fold lower in mutants than in wild-type plants (Table 1). The activation of the OPPP as a possible mechanism to alleviate the blockage caused by the reduced capacity in Suc biosynthesis was already reported in the double-mutant *spsA1*/*spsC* [17]. Interestingly, in mutant *spsa2*, the lack of SPSA2 cannot be circumvented by the previously reported transcriptional overlap of *SPS* genes [15,16,17]. 

The imposition of a slight level of osmotic stress enabled the analysis of the effect of mutation over a prolonged period. In agreement with [18], the transcriptional activation of *SPSA2* in response to mannitol was confirmed (Appendix A). However, the physiological changes in the *spsa2* mutant were observed with some delay after the application of the stress. The total soluble sugars were lower in the mutant than in wild-type plants under control conditions, and the values became similar at 0.5 DAT and then lowered again at 4.5 and 6.5 DAT (Figure 2a,b,d). Considering the overlapping function of SPSs, it is possible that the activation (which could be both transcriptional and post-transcriptional) of other SPS isoforms under stress may have compensated for the lack of SPSA2 at the beginning of the stress experiment. In support of this hypothesis, *SPSC* is known to be activated by osmotic stress, albeit to a lesser extent than *SPSA2* [18]. However, under prolonged stress conditions, the supposed activation of any SPSs transiently substituting *SPSA2* was not sufficient, further supporting the previously suggested role of SPSA2 in stress responses [18]. 

In the absence of SPSA2, lipid peroxidation was higher in the mutant than in wild-type plants (Figure 4) and correlated with a lower accumulation of Pro (Figure 5). Decreased Pro accumulation in the presence of a lower NADP^+^:NADPH ratio seems quite inconsistent since both enzymes in Pro synthesis were shown to be inhibited by a high NADP^+^:NADPH ratio [45,46]. However, Pro accumulation in *spsa2* plants could be affected by the reduced availability of carbon skeletons for its synthesis. Several other mutants involved in carbohydrate metabolism are in fact known to be impaired in Pro accumulation, suggesting a strict relation between carbohydrate metabolism and stress response [34,36]. 

Surprisingly, the oxidative component of G6PD activity started to decline at 4.5 DAT and became equal to that of wild-type plants at 6.5 DAT (Figure 6). The decrease in the redox-regulated (oxidized) component of G6PD activity correlated with the reduction in the gap between wild-type and mutant plants in the NADP^+^:NADPH ratio (Table 1). At the transcriptional level, besides a slight increase in *G6PD6* in response to stress, both in *spsa2* and wild-type plants (Figure 7f), the decline in the expression of *G6PD4* in wild-type plants at 6.5 DAT was particularly evident (Figure 7d). It is possible that protein–protein interactions or other types of regulation based on SPSA2 may trigger this effect, even in the presence of the other three SPSs.

In conclusion, here, we found that SPSA2 influences the carbon flow through the OPPP and plant responses to mild drought stress (i.e., Pro and sugar accumulation and antioxidant response). Despite the overlapping function and expression of *SPSs*, our analysis suggests that SPSA2 is part of a metabolic or regulatory network that links sucrose synthesis with the production of reducing power and pentose phosphates.

## Figures and Tables

**Figure 1 biology-12-00685-f001:**
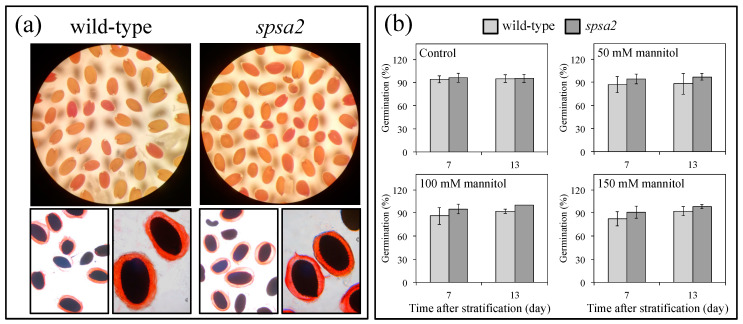
(**a**) TZ (upper panels) and RR (lower panels) staining of wild-type and KO seeds; (**b**) percentage of germination measured in wild-type and *spsa2* seeds evaluated under control and stress conditions. The germination was followed for 13 days after stratification. Data are means ± SDs. A *t*-test was performed, and no significant differences were observed.

**Figure 2 biology-12-00685-f002:**
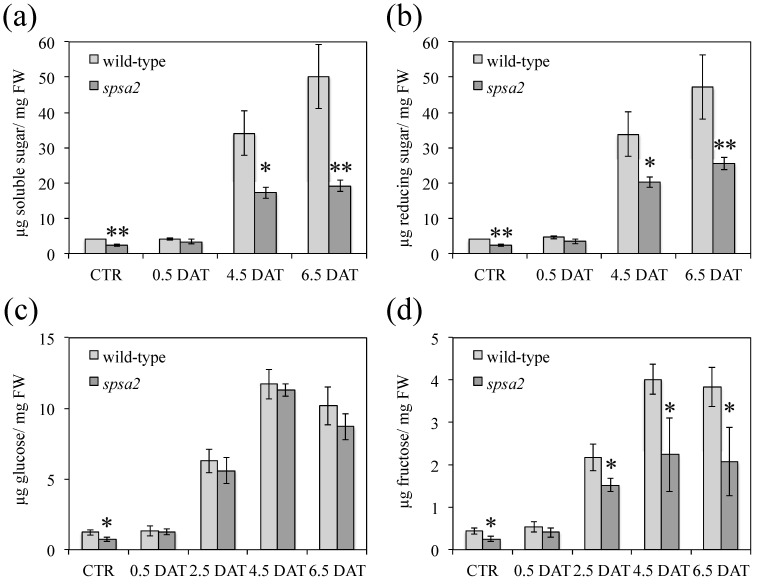
Quantification of leaves (**a**), total WSCs, (**b**) reducing WSCs, (**c**) glucose, and (**d**) fructose pools. Arabidopsis plants were harvested at 12 h light under control (CTR) and osmotic stress conditions (DAT). Three independent biological samples were analyzed for each experimental point. Data are means ± SDs. In pairwise comparisons between genotypes, the *t*-test was used for statistics: * *p* < 0.05; ** *p* < 0.01.

**Figure 3 biology-12-00685-f003:**
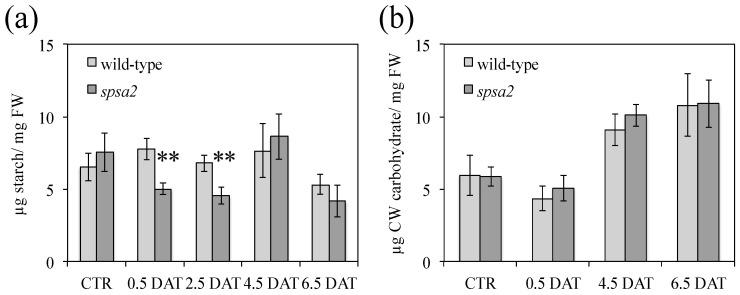
Quantification of water-insoluble carbohydrates in Arabidopsis leaves collected at 12 h light under control and stress conditions: (**a**) starch content in wild-type and *spsa2* plants; (**b**) CW carbohydrates in wild-type and *spsa2* plants. Data are means ± SDs of three independent biological samples. The *t*-test was used for statistics: ** *p* < 0.01.

**Figure 4 biology-12-00685-f004:**
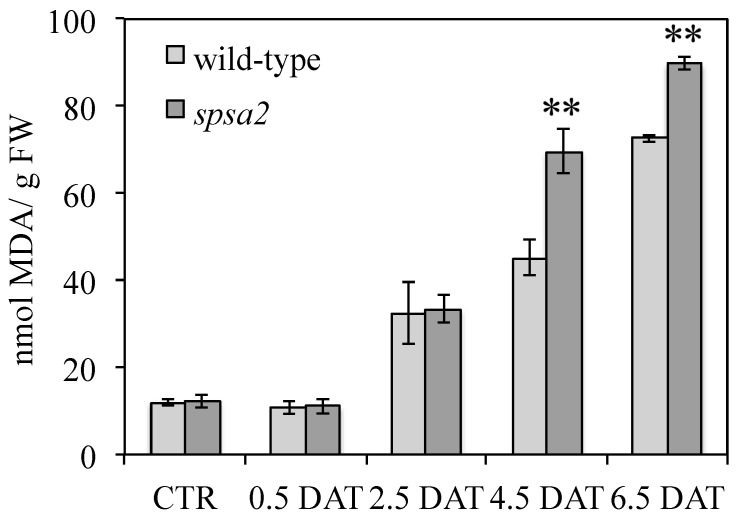
Lipid peroxidation in response to drought. TBA assay was performed on Arabidopsis leaves collected at 12 h light at increasing time after the exposure to 150 mM mannitol. Data are means ± SDs of three independent biological samples. In pairwise comparisons between genotypes, the *t*-test was used for statistics: ** *p* < 0.01.

**Figure 5 biology-12-00685-f005:**
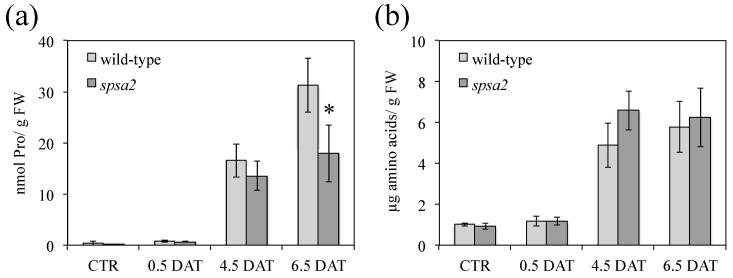
Quantification of Pro and total amino acids in Arabidopsis leaves collected at 12 h light under control and stress conditions: (**a**) free Pro content in wild-type and *spsa2* plants; (**b**) free total amino acid content in wild-type and *spsa2* plants. Data are means ± SDs of three independent biological samples. The *t*-test was used for statistics: * *p* < 0.05.

**Figure 6 biology-12-00685-f006:**
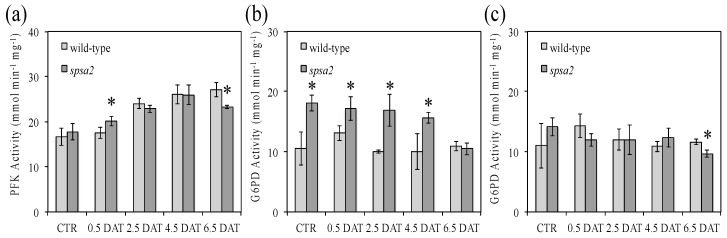
PFK and G6PD activity patterns in response to stress: (**a**) PFK activity measured in wild-type and *spsa2* plants; (**b**) G6PD activity measured under oxidizing conditions in wild-type and *spsa2* plants; (**c**) G6PD activity measured under reducing conditions in wild-type and *spsa2* plants. Data are means ± SDs of three independent biological samples. The *t*-test was used for statistics: * *p* < 0.05.

**Figure 7 biology-12-00685-f007:**
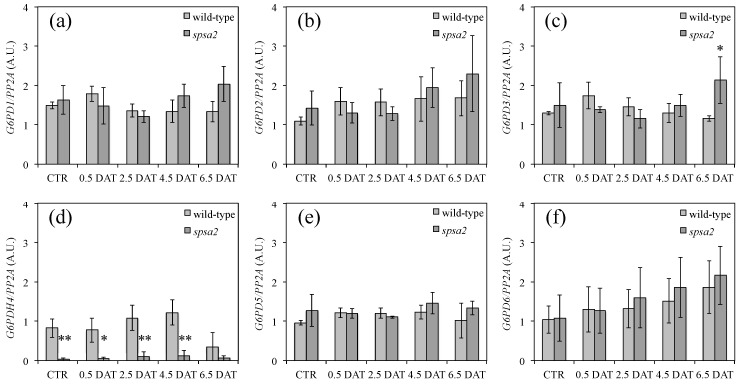
Expression profiles of *G6PD*s under control conditions and in response to drought: (**a**) expression levels of *G6PD1* normalized on the expression level of *PP2A*; (**b**) the same as (**a**) but refers to *G6PD2*; (**c**) the same as (**a**) but refers to *G6PD3*; (**d**) the same as (**a**) but refers to *G6PD4*; (**e**) the same as (**a**) but refers to *G6PD5*; (**f**) the same as (**a**) but refers to *G6PD6*. The intensity of the bands was quantified with ImageJ. Three independent biological replicates were analyzed. Values are reported as means ± SDs. The *t*-test was used for statistics: * *p* < 0.05; ** *p* < 0.01.

**Figure 8 biology-12-00685-f008:**
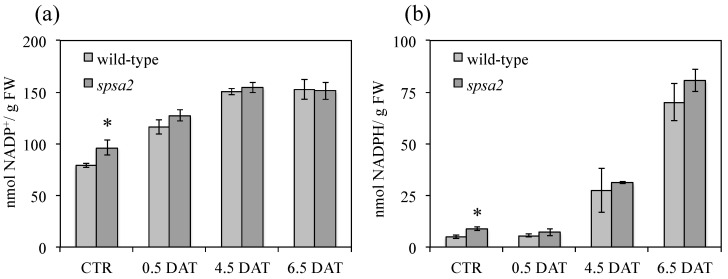
Effect of drought on NADP^+^ and NADPH contents in Arabidopsis leaves: (**a**) NADP^+^ content in wild-type and *spsa2* Arabidopsis plants exposed to 150 mM mannitol; (**b**) the same as in (**a**) but refers to NADPH. Values are reported as means ± SDs of three independent biological samples. The *t*-test was used for statistics: * *p* < 0.05.

**Table 1 biology-12-00685-t001:** NADP^+^:NADPH ratio in *spsa2* mutant and wild-type plants evaluated under control conditions and in response to 150 mM mannitol treatment. Plants were harvested at 12 h light.

	CTR	0.5 DAT	4.5 DAT	6.5 DAT
	(NADP^+^:NADPH)			
Wild-type	15.7:1	20.7:1	5.5:1	2.2:1
*spsa2*	10.7:1	17.9:1	4.9:1	1.9:1

## Data Availability

The data are contained within the article.

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
