# Peer review of "Arabidopsis thaliana* Sucrose Phosphate Synthase A2 Affects Carbon Partitioning and Drought Response"

_biology, 2023, doi:10.3390/biology12050685_

Round 1

Reviewer 1 Report

Some of the presentation should be improved

Author Response

  1. All gene name should be in italic (Ex: L123, L168, L487-488, etc)

Answer: we have thoroughly checked the entire manuscript and amended the errors.

  1. To estimate the expression levels of G6PD isoforms, quantitative PCR approach is more cogent than that of RT-PCR in MM2.9.

Answer: we agree with the reviewer’s comment. A quantitative PCR approach is indeed more cogent for quantitative analyses. In order to limit costs, we opted anyway for an RT-PCR analysis although this method yields only qualitative results. As a consequence, our comments on the G6PDs expression levels are only qualitative, and we carefully avoided any attempt of comparison among the different isoforms between the two genotypes. We believe that this experimental approach gives sufficient information for the goals of our manuscript.

  1. To monitor the transcript of SPSA2 under mannitol treatment, a negative control under normal condition at different DAT is expected.

Answer: the reviewer is right. However, the induction of SPSA2 in response to mannitol/osmotic stress is well documented in the literature. The purpose of the experiment reported in Figure 2 was simply of a confirmatory type. In fact, it does not add any novelty but simply confirms previous data from other groups. Because of the deadline for manuscript resubmission, at the moment we cannot repeat the whole experiment.

  1. The "?" symbol was present at the asterisk mark in all the histogram, why?

Answer: Sorry for the inconvenience, but I assume it depends on a formatting error. I fixed the problem by changing the font of the symbol.

  1. Most importantly, the supplement of results from the functional complement of spsa2 mutants was more convincing.

Answer: We are not sure of which is the exact meaning of the reviewer’s suggestion. Maybe he/she meant “functional complementation”. Functional complementation in the context of elucidating the function(s)/role(s) of a gene was defined as the ability of a particular homologous or orthologous gene to restore a particular mutant with an observable phenotype to the wild-type state when the homologous or orthologous gene is introduced in cis or trans into the mutant background (J Vis Exp. 2016; (112): 53850). We agree that this approach would provide more definitive results, but it would require a tremendous amount of work, and it is clearly not feasible within the short time we have to revise the manuscript. Anyway, we thank the reviewer for this suggestion that we could use for a continuation of the research in the future.

Reviewer 2 Report

In their study "Arabidopsis thaliana sucrose phosphate synthase A2 affects 2 carbon partitioning and drought response", Bagnato et al. present an article of overall very high quality. 

I just have a few remarks regarding the framing of the article.

The authors write "and it has been predicted that all higher plants have at least one member in each" -> Please do not use the term "higher plants", because this is phylogenetically incorrect. A tree can be flipped at any node because all organisms in a given clade are equally divergent from that node (their respective last common ancestor). Thus there can be no higher and lower in a phylogeny. It is also unclear what this means: vascular plants? Land plants (Embryophyta)?

Furthermore, I am not sure that this will hold based on the current knowledge.

(A)

First of all, the closest algal relatives of land plants show expansions in the repertoire of SPS homologs, see and cite:

Cheng S et al. 2019. Genomes of Subaerial Zygnematophyceae Provide Insights into Land Plant Evolution. Cell. 179:1057-1067.e14. doi: 10.1016/j.cell.2019.10.019.

Jiao C et al. 2020. The Penium margaritaceum Genome: Hallmarks of the Origins of Land Plants. Cell. 181:P1097-1111.E12. doi: 10.1016/j.cell.2020.04.019.

Dadras A et al. 2022. Environmental gradients reveal stress hubs predating plant terrestrialization. Evolutionary Biology doi: 10.1101/2022.10.17.512551.

Feng X et al. 2023. Chromosome-level genomes of multicellular algal sisters to land plants illuminate signaling network evolution. Evolutionary Biology doi: 10.1101/2023.01.31.526407.

(B) There is also reported dynamics of sucrose under stress in these organisms, see and cite 

de Vries J et al. 2020. Heat stress response in the closest algal relatives of land plants reveals conserved stress signaling circuits. The Plant Journal. 324:1064–24. doi: 10.1111/tpj.14782.

The authors used a t test. Did they test for normality of the data?

Author Response

I just have a few remarks regarding the framing of the article.

  1. The authors write "and it has been predicted that all higher plants have at least one member in each" -> Please do not use the term "higher plants", because this is phylogenetically incorrect. A tree can be flipped at any node because all organisms in a given clade are equally divergent from that node (their respective last common ancestor). Thus there can be no higher and lower in a phylogeny. It is also unclear what this means: vascular plants? Land plants (Embryophyta)?

Answer: We meant to refer to land plants. The phylogenetic analyses conducted in ref. 15 indeed analysed only land plants. We apologize for the mistake and thank the reviewer for the comment. Considering that the Introduction is not intended to have any evolutionary ambition, we have rewritten the sentence accordingly.

  1. The authors used a t test. Did they test for normality of the data?

Answer: yes, we tested the normality of the data before using the t-test.